# The Art of Childbirth of the Midwives of Al-Andalus: Social Assessment and Legal Implication of Health Assistance in the Cultural Diversity of the 10th–14th Centuries

**DOI:** 10.3390/healthcare11212835

**Published:** 2023-10-27

**Authors:** Blanca Espina-Jerez, Ana María Aguiar-Frías, José Siles-González, Aliete Cunha-Oliveira, Sagrario Gómez-Cantarino

**Affiliations:** 1Department of Nursing, University of Alicante, Carretera de San Vicente del Raspeig s/n, 03690 Alicante, Spain; jose.siles@ua.es; 2ENDOCU Research Group (Nursing, Pain and Care), University of Castilla-La Mancha, 45071 Toledo, Spain; 3S. João de Deus Higher School of Nursing, University of Évora, Largo do Sr. da Pobreza 2B, 7000-811 Évora, Portugal; anafrias@uevora.pt; 4Health Sciences Research Unit: Nursing (UICISA: E), Nursing School of Coimbra (ESEnfC), 3004-011 Coimbra, Portugal; alietecunha@esenfc.pt; 5Center for Interdisciplinary Studies (CEIS20), University of Coimbra, 3000-456 Coimbra, Portugal; 6Faculty of Physiotherapy and Nursing, Toledo Campus, University of Castilla-La Mancha, 45071 Toledo, Spain

**Keywords:** history of nursing, midwifery, women physicians, gender, cultural diversity, transcultural nursing, health, sexuality

## Abstract

(1) Background: The role of Al-Andalus’s women were the result of Arabization and Islamization in Spain. The 10th to the 14th centuries were a time of significant cultural diversity in the region. Female physicians and midwives were important for providing care to women. Despite existing studies, there is still a lack of focused research on the professionalization of these trades, including their requirements, intervention areas, and treatments. (2) Methods: To address this gap, we conducted a scoping review using the dialectical structural model of care (DSMC). Primary medical and legislative sources were used. (3) Results: two kinds of midwife, or qābila, were discovered, along with a woman physician, or ṭabība, who also acted as a midwife. These professions underwent diverse training and fulfilled duties as obstetricians and pediatricians. Midwives were esteemed members of society and were the sole female professionals who needed qualified training. Their performance in the courts was exemplary. Tools for facilitating childbirth and interventions related to female health were discovered in the study. (4) Conclusions: The patriarchal societies suffer from significant inequality in terms of academic training, knowledge transmission, and healthcare provision. Midwives functioned in segregated domestic and legal spaces and were responsible for providing public care to communities from the 10th to 14th centuries.

## 1. Introduction

Throughout history, women have traditionally been responsible for providing informal care within the household. This care has been structured in the domestic setting, with a strong connection between women and their children in biological bonds formed through lactation and upbringing, which are crucial for the survival of offspring. This cultural and social construct has played a significant role in shaping gender roles associated with care [1,2]. The provision of care is shaped by the society and culture where it occurs, as well as its various structures and diverse social, religious, economic, symbolic, gender, and mental factors. Therefore, a historical analysis necessitates an approach that takes into account the different intersecting structures, establishing a dialogue between official health culture and popular or folk culture [3,4]. The historical study of care provided by Andalusian women is of great significance.

The women of al-Andalus, or Andalusian women, emerged from the intricate assimilation of Arab culture (Arabization) and Islam (Islamization) in the Iberian Peninsula (present-day Spain and Portugal) during the Muslim domination of the region [5,6,7]. It can be argued that the majority of these women held a role in providing care in the home [8,9,10]. However, the intricate formation of Andalusian society and culture necessitates a cautious approach towards its interpretation and comprehension.

From the 8th to the 15th centuries, a multitude of peoples and cultures inhabited the Iberian Peninsula. Once the Arabs arrived on the Iberian Peninsula, accompanied by large Berber armies from the recently conquered Maghreb, the territory experienced a continuous ebb and flow of predominantly Christian and Muslim occupation. The Catholic Monarchs declared the Reconquista complete in 1492 [5,11]. There were up to seven social groups, including Arabs, Berbers, al-Mawali, Hispano-Visigoths, Muladis and Mozarabs, Jews, Normans, and slaves [5,12,13,14,15,16,17,18,19,20].

The diversity of religions in Andalusia was influenced by the variety of ethnicities and groups in the region. Islam, Christianity, Zoroastrianism, Judaism, Buddhism, and Manichaeism were all practiced [21]. However, Muslims considered Christians and Jews as “People of the Book”, ahl Al-Kitab, granting them the choice between converting to Islam or preserving their beliefs, practices, and rituals within the limits of the law [18,19].

Consequently, religion and the process of Islamization had a greater impact on shaping Andalusian society through women than Arabization, at least initially [22]. The Islamic religion and its legal framework played a crucial role in shaping the formation of the new society on the peninsula. The intercultural matrimonial institution was regulated by legal documents, which prohibited Muslim women from other countries and regions other than al-Andalus, such as Arabia, the Maghreb, Persia, Egypt, among others, from marrying a Christian or Jewish man. In contrast, Muslim men (Arab or Berber) were permitted to marry Christian or Jewish women, on the condition that they convert to Islam within the subsequent few months. Failure to do so would render the marriage null and void. The Andalusian population was born from intercultural marriages and educated according to Islam from birth [23,24]. Therefore, this study will use the terms “woman” and “Andalusian woman” interchangeably to describe the Islamic women who lived in the territory of al-Andalus during the 10th–14th centuries. These women were the product of the fusion of various cultures and societies.

The study of Andalusian women in history is a recent development, with the earliest collection of research on the topic published in 1989. Historically, the focus of care in al-Andalus centered mainly on institutional and male spheres, particularly the medicine field led by male figures, which received significant recognition at the time [25]. Notably, Muhammad himself regarded it as the second science after religion [26,27,28]. The sources in this field are plentiful, consisting of medical treatises authored directly by notable physicians such as Abulcasis, Averroes, Avenzoar, Ibn Yulyul, and other Latinized Andalusians. These treatises, produced between the 10th and 14th centuries, furnish substantial insights into mentalities, beliefs, norms, values, rites, and related topics. These care practices range from the most scientific to the most popular knowledge, and even manuals with a scientific approach contain popular remedies for ailments [25,29]. In the history of care, the official and professional health culture is differentiated from the invisible history belonging to popular culture that linked the folk culture of legends, traditions, and magical rituals to care [3]. In the case described, some medical treaties combine both types of understanding and provision of care.

In male-dominated medical and institutional settings, the focus was primarily on male healthcare practitioners, with abundant information available on their practice. Conversely, there is a lack of in-depth studies regarding the female healthcare providers in al-Andalus. Lévi-Provençal (1950) [30] emphasizes the importance of female physicians, midwives, and wet nurses in al-Andalus during the caliphal period and highlights their need for training and prerequisite skills. Maya Shatzmiller (1994) [10] also examines this topic.

Aguirre’s (1997) [31] analysis of Ibn Sahl’s fatwa yields evidence of a ṭabība, a highly skilled female physician, who treated female childhood ailments. This indicates that Muslim women tended to work within female-only contexts outside of the family environment. Aguirre (1997) [31] also considers the potential for the ṭabība mentioned in the text to have received education within the domestic or family setting, similar to the majority of Andalusian women with academic training. Aguirre also examines various professional groups devoted to healing, such as “ṭabīb”, “muṭabīb”, or “mutaṭabīb”, including specialists in bloodletting (fāṣidūn or faṣṣādūn) and healers and charlatans. However, Aguirre refrains from speculating whether these male figures represent a comprehensive picture.

The work of Cabanillas (2012) [32] offers insight into the roles of midwives, or qābilas, within healthcare, particularly their aid during childbirth. In the legal-judicial sphere: (1) the midwife verified that a woman was pregnant, which was important for inheritance rights; (2) she testified in the event of a child dying at birth; (3) she also examined a woman slave’s body before she were purchased, and partially determined her price based on this examination.

A more detailed analysis of the qābila is available in the study by Roldán et al. (2014) [33]. The research presents the tutelage system used for training future midwives, which is mainly based on observation and oral transmission due to the difficulty women faced in writing treatises. It also emphasizes the reasons for the involvement of women in assisting childbirth and newborns, as well as their legal-judicial role. The midwife’s role encompasses three fundamental functions: educational, healthcare-related, and legal. Nevertheless, primary documentary sources remain scarce despite the comprehensive nature of the role.

Throughout the study period spanning the 10th to 14th centuries, this topic is addressed through fragmented medical treatises, with only one manual collating different health issues related to obstetrics, gynecology, and pediatrics having been preserved. This is the work of Arib Ibn Sa’id from the 10th century [34]. Other primary sources cover specific topics, highlighting the significance of collecting and interpreting information about the history of women’s role in healthcare, which has been incomplete until now.

Previous research lacks a focused investigation into the professionalization of Andalusian midwives. Specifically, it fails to address the various healthcare activities they performed outside of obstetrics, their approach to childbirth assistance, utilized therapies, requirements for contracted midwives, and other pertinent issues. Thus, the primary aim was to scrutinize the care rendered by midwives in Andalusia, substantiating the significance of their occupation and capacity in the heterogeneous Andalusian society from the 10th to 14th centuries. The ancillary objectives included (1) determining the categories of midwives according to their training and requirements; (2) establishing their social and healthcare impact on the multicultural and patriarchal society of the study period; (3) understanding functions and relevant aspects within the legal framework of the period; (4) examining their care practice regarding women and newborns.

## 2. Materials and Methods

### 2.1. Study Design

A scoping review was undertaken to analyze the care provided by Andalusian midwives during the 10th–14th centuries. Such a review is an indispensable method for determining the impact of a range of publications and studies on the research topic. It allows researchers to form a more focused perspective when assessing, synthesizing, and critiquing primary documents [35].

The dialectical structural model of care (DSMC) was utilized, as it is a suitable approach for investigating the social and cultural history of care, which is intrinsically linked to the sexual and gender division of labor [3,36]. It is of utmost importance to consider the social and cultural aspects related to the care provided by Andalusian midwives between the 10th and 14th centuries for this research.

The DSMC enables the analysis of care structures to establish relationships between them comprehensively. The structures used were (1) functional unit (FU), which represents the beliefs, knowledge, and feelings of people who live and socialize under the same social structure, through which social and cultural systems are constructed that determine the sexual and gender division of labor, so decisive in health and care professions. In this case, it refers to the types of midwives according to their training, the social assessment of midwives, and their regulated medical and legal requirements, which shape the pseudo-professionalization of care provided by women during the 10th–14th centuries. These reflect beliefs, knowledge, norms, and legal and professional limits; (2) functional framework (FF), which refers to the space enabled to carry out care activities, specifically, the domestic area, with a strongly segregated female space, and the legal area; (3) functional element (FE), which integrates social actors responsible for the assistance as well as the care actions they perform, in this research, the areas and forms of physical, psychological, and sexual care, always interlacing a biological and folk component, found through medical manuals and recipes of the time [37,38,39].

The research proposes six interrelated thematic blocks, analyzed using the structures of the DSMC [36] (Figure 1).

### 2.2. Search Strategy

The review started with a research question to amalgamate and critically review the current knowledge base [40,41]. In this case: “What was the impact of the midwife figure at the social and care assistance level in the patriarchal society of al-Andalus during the 10th–14th centuries?”

To answer this inquiry, DSMC was applied. The researchers reached a consensus on eligibility criteria, which encompassed data on the midwife in Al-Andalus in the Middle Ages. The review analyzed the various midwife classifications based on their training, social assessment, representation at the labor level, as well as the medical and legal regulations of their profession, including their locations of intervention. Additionally, the study examined the midwife’s care in terms of gestational and newborn health and women’s and childhood diseases from the 10th to the 14th centuries.

The review examined a range of primary sources including medical and legal treatises, as well as recipe books from the 10th to the 14th centuries. The consulted materials also consisted of peer-reviewed articles, official dissertations, procedures, and reports, while congress abstracts, proposals, and editorials were excluded. The documents consulted were in English, Spanish, and French.

The initial search aimed to establish the research topic’s background and scope. Several databases were consulted during this phase to accomplish this objective: (1) PubMed; (2) Cochrane; (3) bibliographic database on health in Latin America (CUIDEN); (4) Scopus; (5) Web of Science; (6) SciELO; (7) PARES (Spanish Archives Portal); and (8) Centre for Higher Scientific Research (CSIC) Arab Studies Publications Database.

The databases were explored using natural or free-text language, which was normalized and controlled with MeSH and DeCs descriptors. These were combined with Boolean operators (“and”, “or”, “not”). The results obtained alongside the search equations and filters utilized to attain them are presented in Table 1. In order to have up-to-date information, the search was limited to the last 30 years. However, due to the historical nature of the topic and limited bibliographic updates, earlier documents were consulted and selected based on their historical significance.

After an initial search, primary and secondary sources were consulted in both physical and virtual formats from various territories through the University of Castilla-La Mancha Library in Spain. This included the School of Translators Library in Toledo, which contains many primary sources translated from Arabic to Spanish, as well as the Public Library of Toledo. Technical term abbreviations were explained upon initial use. The text is free from grammatical errors, spelling mistakes, and punctuation errors. International libraries consulted included the Escola Superior de Enfermagem San João de Deus library (Évora, Portugal), the General Library of the University of Coimbra (Portugal), and the library of the Escola Superior de Enfermagem de Coimbra (Portugal). Additionally, primary sources were accessed from digital libraries, such as the Digital Hispanic Library and the French National Library (Table 2). To ensure present-day relevance, the search was restricted to the past decade. Nevertheless, considering the topic’s historical significance and bibliographic limitations, prior documents were screened and hand-picked based on their significance.

Furthermore, we applied inclusion and exclusion standards during the documentation review and filtration, as depicted in Figure 2. Overall, we identified 96 documents that adhered to the mentioned criteria.

### 2.3. Data Analysis

The documentary analysis was carried out from a qualitative perspective, following the study objective in a systematic manner. The steps followed in the analysis were (1) a thematic link; (2) a preliminary classification of documents based on the inclusion and exclusion criteria; (3) selection of relevant information; (4) interpretation and comparison of results. The selected material was analyzed from the point of view of the four thematic study blocks, each of which is encompassed in the structures of the DSMC: (1) midwife training; (2) requirements to practice; (3) social assessment of the midwife; (4) segregated domestic area; (5) legal area; (6) physical, psychological, and sexual care. These blocks were contextualized in the Andalusian population of the 10th to 14th centuries.

To extract and summarize the data, the first and second authors carried out a general data extraction. The third author examined the findings in depth, while the fourth author identified the six thematic blocks encompassed in the structures of the DSMC from the functional unit, functional framework, and functional element. Discrepancies were resolved by consensus among the researchers. Thus, after working with all the material, it was possible to answer the initial question of this study: “What impact did the figure of the midwife have on the social and care level in the patriarchal society of al-Andalus from the 10th to 14th centuries?”

## 3. Results

### 3.1. The Midwife Trade: Types and Requirements

The term most used in Arabic to refer to a midwife is qābila, which translates to “the one who receives (the newborn)”. This term is used to differentiate professional midwives from those who possess in-depth experience but lack any formal training or professional qualifications [42]. Frequently, experienced elderly women assist family members and neighbors during and after childbirth without holding professional designations.

The term qābila was used to refer to the professional midwife in Islamic territories until the Mamluk period, from the 13th to 16th centuries, including the territory of al-Ándalus. Consequently, the Andalusian midwife was also known as qābila [8,43]. Following the Catholic Monarchs’ complete reconquest of Spain in the 15th century, she was referred to as obstetric, matron, or midwife, regardless of her Muslim faith [44,45]. In Islamic territories, the qābila was renamed daya after the end of the Mamluk period in the 16th century. Daya is a term that is still used in some Muslim societies to refer to traditional midwives [46].

Medical and legal texts from the Middle Ages in al-Ándalus show that midwifery was a socially differentiated trade for women [10,47]. The qābila, or professional midwife, underwent training by more experienced individuals, often within their own families, and applied a systematic methodology in rendering assistance. The midwife learned her trade via oral and generational transmission, through the tutelage of an experienced qābila. Women in al-Andalus were trained in reading and writing the Qur’an, so they learned to read. It is not clear from the sources whether the existing medical texts on gynecology and obstetrics were accessible to midwives afterward [8]. They provided aid with both standard and intricate deliveries, including those that required the use of instruments, and midwives were remunerated for their services [48].

Two types of midwives were commonly present in al-Ándalus: the qābila, or professional midwife, and volunteer assistants. The qābila usually acquired an empirical education without a structured theoretical basis by learning from another qualified and experienced midwife. During the learning process, she worked as an assistant. Volunteer midwives, who were often elderly relatives, neighbors, or friends, helped pregnant women when a qābila was unavailable or by personal choice [8]. Both the qābila and the assistant could share the delivery room, with the former positioned in front of the pregnant woman and the latter—a relative or a student—behind her, either sitting, kneeling or standing and functioning as “the supporters”. The assistants’ duties entailed massaging, encouraging, and accompanying the mother throughout the delivery process [49].

Nevertheless, there were some rare instances when certain women achieved great proficiency as midwives or physicians. These women were educated by male medical practitioners in their families, who imparted a certain level of theoretical knowledge, with access to medical treatises, and provided them with a more structured education. Women who exclusively practiced medicine were referred to as ṭabība, or woman doctor [8,13,47].

The qābila had a higher public profile than assistants or experienced elderly women, as they were involved in deliveries and also had a legal role in court [47]. Ibn Khaldun, a 14th-century physician and jurist, recognized the urban qābila as skilled obstetricians and pediatricians. He believed that they were highly knowledgeable about the embryo and uterus, justifying their reputable and trustworthy position [42]. However, there is limited information available on the professional qualification frameworks prior to the 14th century.

From the 9th century, literary sources depicting daily life also recognized midwifery as a profession, in addition to medical sources. Technical abbreviations are defined upon first use. According to these sources, husbands from affluent families hired midwives on a permanent basis, providing payment to assist in the delivery of their babies and to collaborate in raising their children. Furthermore, they made a distinction between skilled and unskilled midwives. The sources acknowledged the former as authorized specialists in gynecology, obstetrics, and pediatrics, distinguishing them from the latter, who were elderly women [50].

In medieval Muslim society, similar to Christian Europe, the male physician (ṭabīb) was highly revered with greater medical authority and was associated with theoretical medicine. In contrast, women who were dedicated to the fields of care, medicine, and midwifery were usually experienced practitioners, possessed empirical knowledge, and were often compared to the male practical physician (mutaṭabīb) [51].

Soranus of Ephesus stated that the qābila should be a woman with both theoretical and practical knowledge, equipped with the necessary instruments, experienced, and able to follow hygiene standards, as well as being compassionate and tender. Additionally, the qābila should be prepared for assistance. She was required to possess knowledge on identifying symptoms of an imminent delivery, treating a pregnant woman, distinguishing between normal and complicated deliveries, positioning the mother on a specialized chair, placing her assistants adjacent and behind the mother, and handling complications that might arise, like exceptional and hazardous birthing positions or stillborn fetus extraction. Additionally, Soranus addressed postpartum care for the newborn. The author addressed practical matters concerning newborns, including umbilical cord treatment, initial bathing, and protection practices such as the use of salt [52] and of magical rituals that persists today in some Muslim societies [53].

Arib Ibn Sa’īd emphasized the importance of women in these practices and their requisite expertise, experience, and equipment. In order to examine women, the midwife needed to adhere to a range of hygiene protocols, including ensuring that their nails were short when conducting vaginal examinations, palpating the placenta, and receiving the fetus [34].

### 3.2. The Social Position of the Midwife: Legal Aspects

#### 3.2.1. The Social Assessment of the Andalusian Midwife

The complexity of women’s situation during this historical period is evident. Although Andalusian society was patriarchal in nature, there existed a flexibility in attitudes and norms which led to a certain leniency in gender roles and relationships. Women began to be valued for their intelligence, culture, and knowledge [19,32,54]. Biographical dictionaries were even dedicated to individuals who were considered ‘wise’. In terms of women, these were individuals who committed themselves to both religious and non-religious fields, including poetry and medicine, among others [55].

The initial category of occupations for women’s lives consisted of roles that demanded prior training and abilities. These encompassed occupations such as woman doctor (ṭabība), midwife (qābila), and wet nurse (murḍī’), which were all present in Andalusian Spain. The prominence of midwives is evidenced in Ibn Khaldun’s Muqaddimah, written in the 14th century, which includes a section dedicated to this profession that acknowledges it as the highest in the professional hierarchy [10].

Obstetrics and gynecology emerged as a profession organically to manage challenging or complicated deliveries, with the qābila becoming renowned for providing the best assistance [49].

Women who specialized in caring for others (qābila and ṭabība) provided support to women and children, as well as practicing ophthalmology and other forms of medicine both inside and beyond their families. This led to their sizable contributions to society [56]. Their work took place primarily within the domestic sphere, as they were integrated into family life [8,57].

According to Islamic law, male doctors were prohibited from examining or touching the naked bodies of women, especially in intimate areas. Abulcasis acknowledged that if a skilled female doctor was not available, a suitable alternative would be a eunuch doctor, an experienced midwife in women’s diseases, or a woman who could be trained in this field [58]. Gender-based space segregation was evident in the sexual division of labor. Islamic law mandated that only midwives could tend to women, with exceptions made only when the life of the mother and/or the fetus was at risk [59].

They offered both physical and psychological care, as well as crucial social assistance. They performed tasks both within and outside of the household, playing a vital part in women’s lives as well as in broader society. Additionally, regardless of whether they were officially trained midwives, their use of magical techniques conveyed a distinctive social standing within the female sphere [60]. Their marginal social status, according to patriarchal conventions, afforded them a degree of liberty to navigate public spaces, including at night, and to interact with men, specifically the husbands of women in labor [57]. The women in labor themselves favored midwives for their provision of medical care and emotional aid during and after birth [46,61]. In contrast, the qābila had sufficient credibility to serve as the singular witness in a trial before the earliest Muslim officials [42].

#### 3.2.2. Known Women: Names of Those Dedicated to Care

There are two documented instances of Andalusian female healers who also worked as tribal judges—qābilas. One example is Zaynab, a physician from the Umayyad era who died after 1184. She was the sister of al-Hafid Abu Bakr Ibn Zuhr, a member of the renowned lineage of physicians known as Avenzoar. Sources suggest that she was skilled in surgery and ophthalmology. She excelled in medicine due to her proficiency in treating patients. This led the Almohads to select her for the care of their women, children, and women slaves, alongside seeking her counsel on men’s healthcare. In contrast, al-Hasan bint Ahmad b Abdallah, Zaynab’s niece and the daughter of al-Hafid Abu Bakr Ibn Zuhr, is also cited [13]. In fact, al-Hafid Abu Bakr Ibn Zuhr himself acknowledges her proficiency in gynecology, and they practiced as physicians and midwives for the female patients of the caliph al-Mansur in 12th-century Cordoba [62].

The biographical records of Avila (1989) [13] feature Umm al-Hasan bint al-qadi (14th century) from Loja, who was widely recognized as a respected ṭabība. She acquired medical knowledge from her father and was an expert in the Quran, as well as esteemed as the third-best physician in Al-Andalus.

However, it is unknown which qābila, if any, worked during the period of Al-Andalus under Muslim occupation. Following the establishment of Christian rule in 1492, Muslims, Jews, and Christians coexisted in the same space, but not with equal standing. In terms of providing childbirth assistance, the Ordinances of 1498 banned interracial mixing, explicitly stating that no Muslim woman could assist in the birth of a Christian woman and vice versa [38]. Nevertheless, four documented cases exist of Muslim midwives who provided care to pregnant women belonging to royal families, despite legal restrictions. One of the most prominent cases is that of Leonor de Trastámara, queen consort of Navarre, who received care from a Moorish midwife born in Toledo named Xençi or Exenti during the late 14th and early 15th centuries. In her two preceding deliveries, the queen received aid from Marien, a Muslim woman of unknown origins [63].

In late 14th-century and early 15th-century Toledo, two other Muslim midwives assisted in royal births in addition to those previously mentioned. Socially known as Doña Fátima, one of the midwives assisted Queen Consort of Castile, Doña Catalina de Lancaster, while Fátima’s daughter, named Haxa, was the midwife for Queen Doña Blanca de Navarra [64]. Queen Catalina gave birth to her first child at the age of 28, which was considered an advanced age at that time. Due to the risk associated with childbirth in Illescas in 1401, King Enrique III, the queen’s husband, granted Doña Fátima and Maestre Abdalá, her husband, exemption from the Muslim community of Toledo’s annual tax payment of 700 maravedis, which they had successfully developed. Over time, Fátima and her daughter Haxa provided midwifery support for royal deliveries, including that of Doña Blanca de Navarra, who was the wife of Juan II, son of Enrique III and Catalina de Lancaster. In 1436, Juan II instructed his chief accountants to honor tax exemption for Haxa and her children [45,64,65].

Additionally, a Muslim midwife was discovered in Segovia during the latter half of the 15th century, who aided women from less privileged social classes [64].

#### 3.2.3. Unknown Women: Made Invisible in Medical Treatises

Although the qābila had a significant social standing, written medical discourse was predominantly practiced by male physicians since women were prohibited from writing medical treatises, or kitab [33,66].

In the medical treatises that cover pregnancy, childbirth, and postpartum issues, the involvement of midwives is hardly touched upon and only indirectly referred to. In the 10th century Andalusian treatises of Arib Ibn Sa’id and al-Zahrawi (Abulcasis), it appears that midwives played a prominent role in assisting with childbirth and women’s health issues [34,58]. These doctors’ knowledge of childbirth and female care, despite not providing care themselves, implies their recognition of capable and dependable midwives of that period.

However, this was uncommon. In many medical treatises concerning pregnancy, childbirth, and postpartum, the midwife’s role is typically discussed in an indirect and subordinate manner compared to that of male physicians. The midwife is usually depicted as an assistant to the male physician. She is allowed to examine women’s bodies and apply treatments that involve contact with intimate areas only under the direct supervision and guidance of the male physician [67]. It is contradictory that while midwives had direct knowledge and contact with women, male physicians were only permitted to assist in extreme cases when the life of the mother and/or newborn was at risk [59].

Pormann and Savage-Smith (2007) [56] state that male physicians mainly concentrated on dietary and medicinal treatment, leaving midwives responsible for manual methods.

#### 3.2.4. The Midwife in the Courts: Legal Function and Other Aspects

The midwife’s role as a witness in Muslim societies dates back to the 9th century. Their statements were classified as expert testimony, distinct from other types of testimony [68].

The qābila was summoned to court for two primary reasons: (1) The midwife’s role in court was limited to two areas: the exploration of the reproductive organs, including menstruation, health assessments, sexual competence, virginity at the time of marriage, and pregnancy; (2) moral suspicions, such as spontaneous or induced abortion, fetal age at the time of termination, special circumstances surrounding birth, the first cry of the newborn, and verification of the mother’s identity [42]. Women were excluded from participation in trials concerning disputes over marriage, divorce, or criminal and penal cases [68,69].

The qadi or judge had the authority to call upon one or more midwives to act as expert witnesses and carry out the required physical examinations, providing circumstantial evidence relating to events they had not themselves directly observed [42,70]. The midwives’ testimony was evaluated prior to their employment on potential moral implications, in anticipation of potential court appearances. A standard format for collecting statements from midwives in cases involving the death of a newborn after birth was established by the Andalusian jurist al-Talayfuli, belonging to the Maliki legal school. Furthermore, the jurist stressed that experts must be chosen based on their reputation, impartiality, and integrity [71].

Beyond the legal domain, the qābila was responsible for appraising potential female slaves to determine their worth and marketability [14,33]. Moreover, they provided housing for women prisoners, as there were no designated women’s prisons, and supervised their sentences [19].

Professional midwives were employed and compensated for their services, with occasional exemptions from taxes, as previously mentioned. Nevertheless, the discussion relating to midwife employment centered on determining the contracting party and, consequently, who should cover their fees. The Maliki legal school, prevalent in al-Andalus, ruled that in situations where it is evident that the pregnant woman would require the assistance of a midwife, that is, when the safety of the mother and fetus is at risk, the husband shall be responsible for hiring and paying the midwife. Alternatively, if it is predicted that the expectant mother can manage without a professional midwife’s aid, she will have to take charge of the hiring process [71]. However, in al-Andalus, the law was not consistently followed in practice, as is often the case in similar circumstances. For instance, the Maliki jurist Ibn Ḥabīb (Cordoba, 852) mandated that husbands be responsible for hiring and paying the midwife [72].

#### 3.2.5. Blood Kinship: An Unwritten Law

Similar to the milk kinship between the wet nurse and the nursing infant, which resulted in a consanguineous relationship with corresponding legal and relational implications [37,73], the relationship between the midwife and the other participants present during childbirth was less structured.

A quasi-maternal bond was formed between the midwife and the newborn, which could potentially extend to the midwife’s relatives. However, unlike the wet nurse [71], the symbolic incestuous relationship [1] was not resolved through legislation. Additionally, in Morocco it is believed that a special bond can be formed between the mother and midwife when their blood comes into contact during birth [74].

### 3.3. The Midwife’s Care Role

#### 3.3.1. Care during Pregnancy, Childbirth, and Immediate Postpartum: Mother and Newborn

Midwives were hired by the Abbasid caliphs to diagnose their women’s pregnancies and care for them in the pregnant women’s own homes [75]. Nevertheless, Arib Ibn Sa’id’s and Abulcasis’s treatises recognize more tasks related to pregnancy (Table 3).

Various signs are described for diagnosing pregnancy: (1) Signs of suspicion include perception of fetal movements. (2) Signs of probability include weight in the back and lower abdomen, uterine growth, soft cervix (Godell’s sign), and increased vaginal moisture. (3) Physiological changes include the appearance of varicose veins, increased frequency of urination, chloasma gravidarum and alba line, and more erect breasts with changes in coloration [34].

Arib Ibn Sa’id explains that miscarriages occur after forty days of conception, and the likelihood of their occurrence diminishes thereafter. Furthermore, he recommends that pregnant women should avoid going hungry or fasting during pregnancy, excluding them from the annual Ramadan period [34].

When delivery is imminent, the midwife identifies objective signals such as reported pain in the back and lower abdomen, cervical dilation and softening, and amniotic fluid leakage during a vaginal examination. The patient may also experience an increased frequency of urination. The amniotic fluid may contain bloody content due to premature placental detachment [34]. It is important to note that all evaluations are marked as objective.

When a woman tightens her abdominal muscles, feels the need to breathe more air and experiences contractions, she should walk and sit down slowly and carefully. The midwife will carry out regular vaginal examinations and, once it is deemed necessary to initiate labor, she will seat the patient carefully in a chair or armchair equipped with a notch, allowing the midwife to sit in front of the woman [34]. Abulcasis recommends allowing the natural progression of childbirth to take place, and occasionally exerting pressure on the woman’s abdomen to facilitate a faster birth of the fetus [58]. During delivery, the involvement of three additional women is depicted, one positioned on each side of the woman to provide support, encouragement, and comfort, and another at the back to offer assistance when she leans back. At this stage, the midwife may apply an ointment to ease the fetus’s progression through the birth canal. Once the head emerges, the midwife will cautiously assist the fetus [34].

The qābila should always have their own materials for wrapping the newborn and sanitizing the delivery area, alongside medicinal preparations and instruments (refer to Table 4). Abulcasis advises employing the instruments solely in cases of an extended labor that endangers both mother and fetus.

For administering certain medicinal treatments, a vaginal vaporizer in a funnel shape was employed. The narrow end was inserted into the vagina, and the wide end was placed close to the boiled preparation. The vapors were allowed to release until they diminished. Table 4 provides a list of most commonly used medicinal preparations for childbirth and other female remedies.

After delivery, a cloth is placed on the infant to deter cold exposure. The umbilical cord is severed by measuring four fingers from the child and always when it starts to cry. The cut should be made with any material except for iron, as it was once believed to be a bad omen [58]. Subsequently, it is secured with a fine piece of fabric. Following this, the infant is bathed in saltwater, and once dried, salt is applied all over the body [34]. This is an ancient practice that is still utilized in some Muslim communities today [53]. Eyedrops are administered to the newborn, although the formulation is not specified; it may be similar to current ocular prophylaxis practices. Simultaneously, various preparations for expelling the placenta are described in Table 5. The woman’s genitals are then washed with boiled fenugreek, and she is advised to rest [34].

No references to the performance of a caesarean section have been found within the medical treatises of al-Ándalus [78]. However, Jewish physician Maimonides (12th–13th centuries) does describe the possibility of a fetus being born via an abdominal incision, with some cases even resulting in the survival of the woman [79]. Conversely, within the Christian world, the caesarean section is depicted in the Cántigas de Santa María. This scene is exceptional, as it depicts an intervention occurring while the mother is still alive [38,80].

If a stillborn fetus is present, labor can be triggered through medication. Fumigation or smoking of a preparation containing myrrh, parsnip, opopanax, and sulfur is recommended [34]. Instruments listed in Table 4 may also be used to assist.

Although midwives are skilled in pediatric care, according to Ibn Khaldun (14th century), the presence of a male physician becomes more necessary after a child is weaned [47]. However, physicians were not the only ones concerned with proper infant care. Religious circles also disseminated popularized advice [81].

#### 3.3.2. Postpartum Rituals, Fertility, Contraception, Abortion, and Other Areas of Qābila Intervention

The work of midwives was not purely professional but deeply rooted in social and community contexts. The midwife’s duties encompassed aiding in pregnancy and labor while additionally ensuring the safety and health of the newborn, whether under the mother’s care or that of a wet nurse. To assess the proper progression of breastfeeding and early care, the midwife regularly visited the patient’s home [37].

The qābila furthermore fulfilled other functions within the female socio-sanitary system. She was responsible for performing ear piercings, providing assistance during wedding ceremonies, contributing to female circumcision procedures, coordinating the celebration of subu on the seventh day after a baby’s birth [82], as well as managing other female health conditions (refer to Table 6).

The subu ceremony represented a significant milestone for newborns who exceeded their first week of life, signifying the conclusion of the most precarious maternal period. This event included a first haircut, symbolizing the separation from the intrauterine environment and personalizing the newborn with a name, integrating them into society. Women held segregated celebrations on the seventh day. Objects symbolizing the newborn’s future were positioned near the baby’s head, including the Quran, a tablet with an inkwell and a cane representing their destiny, and a loaf of bread with sugar. The following day, the objects were distributed among attendees. The festivities culminated in a procession at the mother’s house with family and friends in attendance, led by the midwife who carried the baby alongside the mother [57].

Similarly, the knife utilized to sever the umbilical cord persisted beside the baby’s head for 40 days, provided that the mother was alert and supervisory [57].

Furthermore, the qābila’s function seems correlated with wedding festivities. They were liable for dyeing the future wife’s hands and feet with henna and aiding in the preparation of the “defloration” ceremony—the passage to sexual intercourse [46].

Andalusian midwives, like their counterparts in medieval and modern Europe, were regularly consulted on matters concerning women’s sexual health, including conception, contraception, and occasionally, induced abortion. References [83,84,85] provide further support for these findings.

Fertility was a special focus of physicians who authored chapters on gynecology and obstetrics. The social, economic, and legal status of women and their husbands in the medieval Muslim world were determined based on their ability to procreate. Arib Ibn Sa’īd suggested using a wool pessary for conception, placing it at the vaginal level, and dipping it in a mixture of saffron, lavender, bugloss, mastic, balm of Judea, lily oil, and oily fats. In addition, he gathered two additional pessaries, two medicinal prescriptions, and incense to assist with conception [34]. Meanwhile, Ibn Wafid (11th century) listed seven concoctions in his recipe book to enhance sexual desire, five of which were beneficial during intercourse, and one aimed to achieve penile erection [77].

Concerning birth control, medieval Muslim jurists, cognizant of economic worries in extensive households, aimed to approve the practice. In pursuit of their aim, they concentrated on the male contraceptive method of coitus interruptus and neglected female contraceptives like vaginal suppositories, for which midwives held a significant role in their recommendation [86].

In terms of abortion, the Maliki jurists demonstrated a strong stance against it [86]. Experienced and trustworthy midwives were employed to identify cases of voluntary abortion or miscarriage in court. Compensation fixed in case of miscarriage was dependent on their assessments, if applicable [71]. Medical sources usually describe abortion in the context of miscarriage [78,87]. Arib Ibn Sa’īd advised taking particular care during the first three months of pregnancy, as this is when the greatest risk of miscarriage exists. The cited reasons include maternal trauma, extreme fatigue, emotional distress, uterine weakness, reduced fetal nutrition, or vaginal bleeding [34]. As a result, pregnant women have the option to postpone fasting during Ramadan.

In cases where the mother’s life is at serious risk during delivery, abortion may be considered [58,79]. The extraction process will be conducted with the specified equipment and medicinal products (refer to Table 4 and Table 5).

Remarkably, Ibn al-Jatib, a physician who wrote the “Book of Hygiene” or “Amal man tabba li-man habba” in the late 14th century, introduced innovative approaches to enabling the prevention of conception and inducing abortion for women at risk of death during childbirth owing to their narrow birth canal or congenital deformity [88].

In addition to the midwife’s interventions in the physical-biological domain, the qābila recommended superstitious treatments such as attending another woman’s delivery and sitting or passing over the placenta [82].

The utilization of mystical techniques bestowed the midwife with a social standing within the female sphere. Healing through physical means was frequently associated with the practice of rituals or magical instruments, similar to the approach taken by Christian midwives and healers during the Modern Age, who inherited this knowledge [83]. Amulets and other enchanted items were employed to safeguard women giving birth and their infants during and after delivery [85]. The mother could wear them, either revealed or concealed, or hang them in her bed or elsewhere in the house. Amulets could also be administered as medicine by immersing the talisman in a liquid preparation which was then ingested by the mother. It was not uncommon for talismans to feature portions of the Quran either written or spelled out on them [28,60,89].

Ibn Habīb (9th century) as well as Ibn Khaldun (14th century) documented several protective measures against the evil eye for both the mother and newborn. Some of these practices endured into the 20th century, as attested by Fanny Davis in 1986 in her extensive study [90] and also at present [91,92,93]. For instance, the baby was wrapped in blue cloth and given a hat adorned with pearls and one or two gold coins. Alternatively, a bunch of garlic was used, always accompanied by one or two verses of the Quran written on blue fabric. The mother could also place a Quran in a bag at the head of her bed and put an onion, several garlic cloves, and blue beads at her feet. Following the delivery, once the guests had left, the midwife would fumigate the room against the evil eye [27,47,90].

## 4. Discussion

As noted in the introduction, research on Andalusian women is a relatively new field, and there is currently a lack of extensive studies on the role and care provided by these women.

Aguirre’s (1997) [31] research indicates that the ṭabība, or female physician, was typically trained through family lineage and domestic education. With regards to the training of the qābila, Espina-Jerez et al.’s (2019) study [8] lacks clarity in differentiating the experienced elderly women from the professional midwives who were considered professionals in childbirth assistance. It should be noted that the term qābila was solely used for the latter group. This distinction is significant not solely in relation to academic education, but also due to the range of roles that the qābila undertook in contrast to assistants or experienced elderly women, regardless of the possibility of coexisting in delivery scenarios. Moreover, their public presentation differed from each other, as they held a profession that was acknowledged for its social value and care.

According to the findings of the study, midwifery has historically coexisted alongside non-professional assistance provided through informal family networks and voluntary aid. Experienced non-professional women still play a role in supporting childbirth, both in rural and nomadic communities and in urban areas with low-income populations, as well as among women who opt for family support during labor and delivery. The research conducted by Makhlouf-Obermeyer (2000) [74] indicates that this situation persists to the present day. Generational care practices based on ancestral magical beliefs, such as washing the newborn’s body with salt, are still being employed [52,94].

The research aligns with the three functions of the qābila described by Cabanillas (2012) [32]: (1) teaching, which has already been explored in the study conducted by Espina-Jerez et al. (2019) [8]; (2) assistance; and (3) legal-judicial. However, according to Jiménez-Roldán et al.’s (2014) study [33], the functions of midwives are described in a general manner. Specifically, midwives have a role in dealing with childbirth-related issues in the area of assistance and diagnosing pregnancy, providing testimony in the case of fetal deaths during childbirth, and evaluating the physical condition of female slaves prior to purchase in the legal-judicial field. However, these functions lack ratification through primary sources. By contrast, this study showcases the involvement of the qābila in legal proceedings that encompass a wider range of topics. These include matters that pertain to the female reproductive system, several ethical suspicions, disputes, and criminal cases.

The social evaluation of women who are devoted to caregiving has been extensively researched, with references to them being labeled as “wise” in biographical dictionaries [13]. Primary sources [34,47,58] provide evidence for this subject, which has been studied in depth [10,32,49,56,95].

However, the analyzed studies fail to present a specific approach to the midwife’s requirements, her various intervention areas, and methods of operation. The information provided is general and segregated [9,10,31,33,56].

However, no research has examined the drugs and instruments employed during healthcare interventions. This has significant relevance when considering present-day practices. The use of such instruments was restricted to high-risk and lengthy childbirth scenarios involving the mother and fetus. Additionally, a predecessor to the contemporary forceps existed, although its use was restricted to the extraction of stillborn fetuses. A form of vaginal speculum, which improved the visibility of the birth canal, was already utilized. There were also spatulas available for cases where the fetus was stuck to the uterine walls. In addition, scalpels and vaginal vaporizers were used to administer vaginal medications. Nonetheless, oral preparations were also utilized. Irrespective of the method of administration, medical treatises document for midwives the procedures for managing symptoms of pregnancy, preventing abortion, alleviating the pain associated with abortion and childbirth, inducing childbirth, aiding delivery of the fetus in challenging positions or maternal exhaustion scenarios, controlling bleeding, and initiating detachment of the placenta [34,58,76,77].

However, despite the existence of these procedures and materials as represented in manuals, it remains uncertain to what extent they were utilized in practice. It is evident that the proficiency in intervening in normal and complicated deliveries mentioned in sources, as well as the diverse array of materials and medications employed, cannot be found in current studies on care providers from the Modern Age (15th–18th centuries) [83]. Despite Christian laws prohibiting the mixing of races in childbirth assistance [38], the qābila, renowned for their expertise, still provided support to Christian women, including members of royalty such as Doña Fátima and her daughter Haxa, as detailed in this article.

Their ability to intervene was only permitted until the point where the mother and/or newborn’s life was at severe risk, prompting involvement from a male doctor [59]. Nonetheless, despite legal and religious regulations, due to reservations surrounding the examination of women’s intimate areas, male doctors were required to possess significant knowledge but limited practical skills to conduct these obstetric interventions, given their unusual nature. Therefore, the resolution of fetal death cases was most likely delegated to either the ṭabība or qābila, resulting in their subsequent court appearance.

Physical care, provided by the qābila and medicine in general, was imbued with superstitions and rituals in a natural way. Scientific care and folk traditions always coexisted [27,28]. Notably, these women were not judged in the same way as their heirs were in later times [83]. An instance illustrating the significance of avoiding a knife or scalpel made of iron to cut the umbilical cord is its association with a “bad omen” [58].

The qābila took part in postpartum customs. Talismans and traditions were in use to shield the mother and her newborn from the evil eye, which are still in practice today [90]. It is important to note that at the conclusion of the subu ceremony, women would proceed with the baby, with the midwife carrying the child. This practice is reminiscent of the tradition in Christianity where midwives who participated in baptism were called “god-mothers” [38].

Despite the many roles acquired by women caregivers during the 10th-14th centuries in al-Andalus, there was thereafter a slower advancement of women caregivers in Muslim societies than in European societies. For example, it is agreed that the first Arabic-language book written for the training of midwives by a woman was written by Khalila bint Salih in the mid-19th century, translated as “The right guidance in the work of midwifery” [71]. All medical writings relating to gynecology and obstetrics were written by men earlier. However, in the medieval period in Europe, the works of Trotula of Salerno (11th century), the published writings of Louise de Bourgeois (16th–17th centuries), and the treatise written by the English midwife Jane Sharp (17th century) are already known [96].

The primary limitations of this study relate to the unavailability of primary sources specifically focused on the treatment of pregnancy and postpartum care, as well as the aid provided for women’s issues. Additionally, primary sources from this time are predominantly written in Arabic and housed in hard-to-access libraries across numerous countries.

Biographical dictionaries often feature women classified as “wise” due to their social status as members of prominent families, including those who were daughters or slaves of caliphs, rather than for their scientific achievements. Such biases pose significant challenges when constructing the historical perspective of Andalusian women.

Regarding medical, legal, and chronicle sources, ṭabība and qābila generally belong to the urban sphere. For a more comprehensive study and taking into account women dedicated to care in rural areas, it may be convenient to resort to archaeological sources or other material sources, which currently have less momentum for the study of Andalusian women.

## 5. Conclusions

In contrast to the high social and professional value that these women achieved, their appearance in the sources is generally anonymous. In this study, some ṭabība and qābila are presented with their names and surnames. However, such personalization is exceptional, occurring when they are related to academically distinguished relatives or work for royalty, as evidenced. The novelty and individualization of this information can create uncertainties regarding the generalizability of the findings in the absence of corroborating evidence.

It is noteworthy that men, despite their gender, authored literature on women’s healthcare. Women acquired medical knowledge through oral tradition and experience but were prohibited from producing medical treaties. This emphasizes the inherent inequality in patriarchal societies and the lack of educational opportunities for women, as only those who belonged to prestigious families with intellectual and/or economic status were able to obtain the title of ṭabība. Furthermore, medical sources fail to reflect the contributions of Andalusian women who dedicated themselves to nursing. This questions the exclusive objectivity of medical sources when compared to other forms of material like fatwas and chronicles.

It is noteworthy that, during this period, female doctors and midwives were highly regarded as the most esteemed and accepted professions in Islamic society in general. Moreover, they were compensated for their services and, in some cases, even granted tax exemptions.

Due to their profession, midwives held a middle position within a patriarchal society in which women were divided. This separation of spaces led to the creation of tasks for women and daughters, which were linked to classical symbols and values associated with the concept of “motherhood”.

Women devoted to caring would relocate from their own residence to that of another woman to meet the demands for care in a particular population. They were not bound by rigid gender norms like other women, as they were able to move without constraints in public areas, even at night, and engage with male family members within the context of administering care and engaging in court activities. However, through their mediation between the male and female domains, they also upheld the principles of sexual segregation and the boundaries of female territory, greatly cherished by Muslim society, for other women. They functioned as publicly appointed care agents in the private family realm, granting them a unique legal status.

## Figures and Tables

**Figure 1 healthcare-11-02835-f001:**
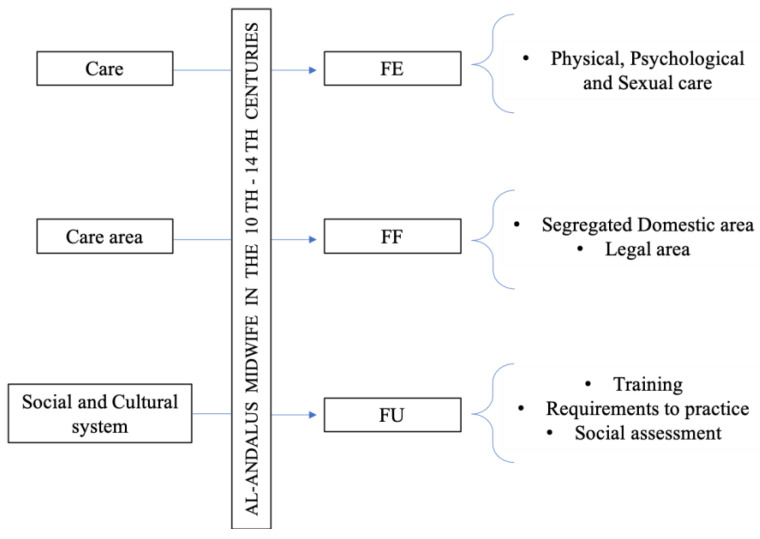
Theoretical dialectical structural model of care (DSMC): application of its structures. Source: authors’ own elaboration.

**Figure 2 healthcare-11-02835-f002:**
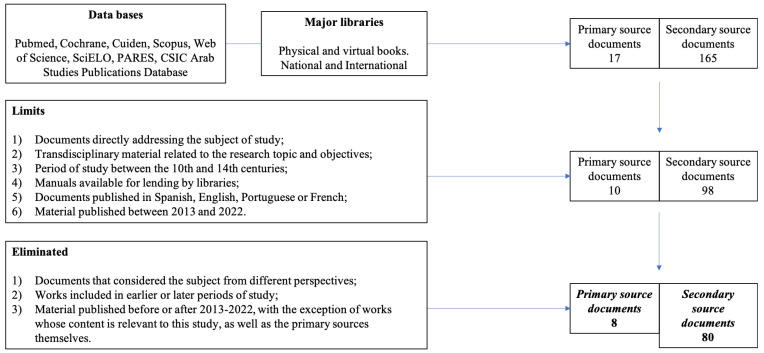
Filtering strategy. Source: authors’ own elaboration.

**Table 1 healthcare-11-02835-t001:** Thematic blocks related to search equations.

Database	Search Strategy	Filters	Points Extracted
**PubMed** **Cochrane** **Cuiden** **Scopus** **Web of Science** **SciELO** **PARES** **CSIC Arab Studies Publications Database**	history of nursing AND midwives nursing AND midwife midwife AND requirements AND Muslim history women doctor AND Muslim history history of nursing AND gender AND legislation midwife AND traditional medicine midwife AND Quran Middle Ages and nursing	Last 10 years Article English/Spanish	Training Requirements to practice Social assessment Segregated domestic area Legal area Physical, psychological, and sexual care

Source: authors’ own elaboration.

**Table 2 healthcare-11-02835-t002:** Primary sources and location.

Primary Source	Location	Source
Ibn Ḥabīb. *Mujtasar fi l-tibb* **(Compendium of Medicine).** 9th century.	Library of the School of Translators in Toledo (Spain)	Al-Ándalus
**Ibn Sa’īd ‘Arīb. *El libro de la generación del feto, el tratamiento de las mujeres embarazadas y de los recién nacidos*.** 10th century.	Loaned by the National Library of Spain through the Library of the University of Castilla-La Mancha (Toledo, Spain)	
**Abu al-Qasim Khalaf ibn al-Abbas Al-Zahrawi [Abulcasis].** *Kitab at-Tasrif.* 10th century.	French National Library	
**Ibn Habib A al M. *Kitab al-Wadiha* (legal treatise).** 11th century.	Library of the School of Translators in Toledo (Spain)	
**Ibn Wāfid. *El libro de la Almohada*.** 11th century.	Library of the School of Translators in Toledo (Spain)	
Abu-l-Walid Muhammad ibn Ahmad ibn Rushd (Averroes). *Kitab al-Kulliyyât al-Tibb (Colliget).* 12th century.	Digital Hispanic Library	
**Ibn al-Jatib M b. A. *Kitab al-Wusul li-hifz al-sihha fi-l-fusul, “Libro de Higiene”.*** 12th century.	Loaned by the Autonomous University of Madrid through the Library of the University of Castilla-La Mancha (Toledo, Spain)	
Ibn Khaldûn. *The Muqaddimah: An Introduction to History.* 14th century.	JSTOR	

Source: authors’ own elaboration.

**Table 3 healthcare-11-02835-t003:** Tasks performed by the qābila according to Ibn Sa’id’s treatise.

Activity	Considerations
Assistance in conception	
Diagnosis of pregnancy	
Normal delivery	9 lunar months When the head comes out before the body, it is easier and safer for the fetus
Premature delivery	7 and 8 lunar months Before that, their survival is not assured
Complicated delivery	Pelvic or breech presentation, transverse position
	Twin births, Siamese twin or multiple fetuses
Deformities of newborns	Greater or lesser number of limbs and/or fingers on hands or feet
Placental extraction	
Hydatidiform mole or molar pregnancy	

Source: Authors’ own elaboration through [34,58].

**Table 4 healthcare-11-02835-t004:** Instruments used by the midwife according to Abulcasis.

Instrument	Use
Vaginal speculum	To open the vagina at its entrance to the uterus. Place the woman in a lithotomy position (“on a sofa with her legs falling and separated”) and insert the lubricated speculum blades. It was made of ebony or boxwood.
Impeller *Midfa’*	There is no clear description of its use. It is interpreted that it could be used to grab the neck of the fetus and thus facilitate the exit of the rest of the body.
Cephalotribe *Mishdakh* This would be the precursor to the current forceps.	It has the shape of scissors whose blades are toothed jaws. It was used to grab the fetus’s head when it could not progress through the pelvic canal, either due to the size of its head (e.g., hydrocephalus is referred to) or the narrowness of the birth canal, and there was no other option for its delivery. It was not intended for the extraction of a live fetus.
Hook or separator *Sinnara* This would be similar to the current spatula.	It has the shape of a bar with a hook at its end. It was used when the fetus was stuck to the walls of the canal to assist in its extraction. Its use is described only in the case that the fetus was already dead.
Perforator or scalpel *Mibda*	It is a kind of scalpel without a handle. It is said that they were designed to be hidden between the midwife’s fingers and not to be seen.

Source: Authors’ own elaboration based on [58].

**Table 5 healthcare-11-02835-t005:** Medicinal preparations used around gestation and childbirth and for other female health problems.

Remedy	Use
Seeds of colocynth boiled in lily oil. Enema.	Prevent miscarriage in the 2nd–3rd month.
Fenugreek, fennel seeds and root, and celery.	Pain from miscarriage. Boil and drink several times a day.
Ginger, musk, celery seeds, asarum, cardamom, nutmeg, cinnamon. Put unperforated pearls of coral or amber. Electuary.	Flatulence during pregnancy.
Mucilage of fenugreek with sesame oil.	Induction of labor during prolonged labor and to increase the quality and quantity of breast milk after 2–3 days of birth.
Pyrethrum or soapwort.	Makes the woman sneeze and thereby contributes to the contractions, for the delivery of the fetus and placenta.
Mucilage of marshmallow, fenugreek, sesame oil, and dissolved gum.	Facilitate the delivery of the fetus in breech presentation, when one of the hands comes out first, or when the fetus has died in the womb. The four ingredients are mashed and mixed, applied to the vulva and lower abdomen. Then, the vaporizer is placed with only clean boiled water. Afterward, combine with pyrethrum. When the placenta does not come out with the previous remedy, apply manually on the placenta inside the uterus, let it act, and pull gently to avoid prolapse.
Pennyroyal, rue, aniseed, chamomile, artemisia, cassia, and centaury. Boil in water and apply with a vaporizer. Combine with pyrethrum. Cooked fenugreek over the genitals once the placenta has been expelled.	Stimulate detachment of the placenta in case of retention or for significant menstrual delays.
Chamomile, myrrh, valerian, carrot seed, anise, fennel seed and root, bitter, sweet, and cultivated rue and juniper. Drink.	Induce menstruation.
Yellow amber, almond gum, pomegranate flower, incense, rose leaves, and nut. Crush with powdered zargatona mucilage and sumac cooking water. Crushed jasmine flowers. Drink.	Stop menstruation when it exceeds the normal duration and any type of vaginal bleeding.
Vitriol (sulphate present in several metals) Quince syrup, toasted gum Arabic, and clay. Drink.	Stop bleeding.
Acacia “Olibanum” or oil of Lebanon Always mixed with egg white.	Stop bleeding and treat pain.
Sumac, pomegranate peel, and oak galls. Administer with a vaporizer.	Persistent vaginal bleeding.
Radish seed ointment with human or sheep milk. Mustard mashed with figs. Among others.	Freckles of the pregnant woman. They will be treated after delivery.

Source: Authors’ own elaboration based on [34,58,76,77].

**Table 6 healthcare-11-02835-t006:** Other areas of intervention by the midwife.

Intervention Cause	Mode
Hermaphroditism	Three possible forms are exposed. In all of them, “what grows must be cut and destroyed”. Then, the usual treatment for wound healing was used.
Excision of the clitoris and other growths in female genitals (female circumcision)	When the clitoris grows above its normal size, and can even become erect like male organs, it must be cut to the root of its growth to prevent serious bleeding. Then, heal and bandage like any other wound.
Imperforate vulva	The woman’s vulva is partially or completely closed. It can be congenital or acquired. If it is congenital, it would be an imperforate hymen. When the membrane is thin, the midwife performs a manual rupture. If it is thick, an incision with a scalpel must be made. Daily care and bandaging with dry linen. Sometimes it is due to large growths caused by cancerous tumors. These cases cannot be remedied by isolating the tumor.
Hemorrhoids, warts, and red pustules in the female vulva	The treatment and prognosis depend on their depth. The shallower they are, the better the prognosis and the easier to cure. The procedure consists of their removal, holding the warts with rough fabric and cutting them with a scalpel. Use medications to stop bleeding.
Uterine eruptions	Collections of pus occur, which must be opened with a scalpel and drained. When acute pain symptoms occur, pulse, inflammation, heat, and redness, use ointments that help to naturally suppurate it before breaking it.
Imperforate anus	Newborns are sometimes born with the anus closed by a thin membrane. The midwife will perforate it with a finger or scalpel, taking care not to reach the muscle. Soak a cotton ball and linen in oil, and treat it after each bowel movement and until it heals.

Source: Own elaboration by consulting primary sources [37,82].

## Data Availability

All data generated or analyzed during this study are included in this manuscript. The range of materials, including primary sources, is cited.

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
