# Peer review of "The Art of Childbirth of the Midwives of Al-Andalus: Social Assessment and Legal Implication of Health Assistance in the Cultural Diversity of the 10th–14th Centuries"

_healthcare, 2023, doi:10.3390/healthcare11212835_

Round 1
Reviewer 1 Report
Thank you for the opportunity to review this article on the history of women of al-Andalus between the 10th and 14th centuries.
My review centers around the historical methods since this is an historical article.
Primary Sources
My major concern is around the use of sources. The authors note that the objective of the article was to analyze the care provided by Andalusian midwives between the 10th and 14th centuries. The authors state the scoping review revealed 92 documents, primarily limited to the past 10 years. It is not clear if they also consulted archives or which archives they consulted? In historical research, sources are divided into primary and secondary sources. Primary sources are documents written at the time of the event (i.e. 10-14th C), and form the bulk of evidence when making historical arguments. I cannot figure out from this article what information came from primary sources and what information came from secondary sources. And I was confused by the authors use of the term primary source throughout. In the graphic on page 6, the authors identify they found primary source documents in including virtual and physical books and also databases, but they do not cite archival collections. I do not understand what the authors mean by primary sources in this article. The authors cannot make a historical argument about the role of midwives in the 10th-14th centuries (their objective) without actually consulting sources from the 10th-14th centuries.
Secondary Sources
This brings me to my next point which is the use of secondary sources (published academic articles and books). Unlike nursing research, historians do not limit themselves to publications in the past 10 years as historical scholarship does not fall out of date the same way nursing research does. Are there key historical works that are missing from this article as a result? The authors seem to draw on published works from a variety of contexts and geographical situations without any description, analysis, or awareness for the variations in geography, locale, context. For example, when the authors note that "two types of midwives were widely found" they do not indicate where? In published works about al-Andalus? In literature pertaining to the entire Middle East? The history of midwifery has taken different forms in different context, and it varied globally. Looking at published works about midwives globally from the 10th-14th century will not allow the authors to answer the question about midwives specifically in al-Andalus in the 10th to 14th century. For example, the authors cite: Goldsmith J. Childbirth Wisdom: From the World’s Oldest Societies, which is not specific to al-Andalus and needs much more attention to specific context. There have been many histories of midwifery that the authors might want to consider reading and engaging with. For example, Hibba Abugideiri's book (2016) Gender and the Making of Modern Medicine in Colonial Egypt , as a start. The authors may want to consider consulting JSTOR also since it will produce more history research results. Again, the secondary sources are not used with specificity and attention to context. For example, the first sentence under results discusses a common Arabic word that defines the role of the midwife and the reference is for a book on expert witness in the Islamic courts. Is this book specific to the geographical region? It is extremely challenging to figure out what is specific to the region under study, and what is more general research about other locales that may or may not be specific to al-Andalus. The authors then go on to say daya is a term “still used in the Middle East to refer to the traditional midwife” but they cite a book about Egypt. The “Middle East” is not a homogenous, monolithic culture. There is not enough specificity in this article to make claims about the specific region under study. In another example, the authors discuss the “evil eye.” In the literature more broadly, this has been addressed and there are many ways that this was observed. This article does not include any analysis. How did midwives in al-Andalus incorporate spiritual beliefs into their practices? Also this practice still happens today in some places in the world, yet the authors cite Fanny Davis in 1986 as confirming these practices persisted into the 20th C. I do not think the authors are aware of the extent and evolution of this practice.
Generalizations
Likewise, when talking about midwives in the court, the authors cite a reference that deals with Syria/Palestine. There is no monolithic "Muslim world or “Middle East”. The authors might want to bring more specificity to their analysis.
Also, wording about religions as "cults" (page 2, line 64) are worth reconsidering. And "Muslim women who came from the East" (line 2 page 73) is unclear and seems othering. How do the authors define and conceptualize "the East"?
I appreciate some of the detail the authors found about midwives in published literature.
There are some slight revisions needed.
Author Response
Thank you for your review of the article.
We believe that there is certainly a lack of clarity regarding the primary and secondary sources, so we have included table 2 which lists the 8 primary sources used, their source of provenance (all from al-Andalus) and the location where they were found (libraries in physical and online format). Some of these libraries contain archival material and, fortunately, many of them are digitised.
In the references, some of these primary sources have more recent dates. This is because this is the date at which they have been translated. During the late 20th century, Arabic studies experienced a great impulse in Spain and many manuscripts from al-Andalus were translated into Spanish, but after that no more work was done on these sources than on their translation.
The search period has also been corrected to 30 years. Initially, the search was limited to 10 years, but as explained in the text, this parameter was withdrawn as works prior to this period had to be included because of their historical relevance to the research.
On the other hand, we would also like to point out that much of the documentation has been extracted from JSTOR and we had not forgotten to include it, so it has been added to table 1.
We would like it to be taken into account that primary sources on this subject are practically scarce, which is why we resort to other secondary sources that are not specific to the territory, which, due to the thematic similarity and, on most occasions, their direct reference to al-Andalus, make us consider them.
Examples of how midwives in al-Andalus incorporated spiritual and popular beliefs into their practices are given in Section 3.3.2. Postpartum Rituals, Fertility, Contraception, and Abortion. Other areas of inter-vention of the qābila.
It is true that the imprint of the "evil eye" is still in the present, and we had not been aware of the importance of providing updated information in this regard. Three works carried out in recent years have been included that reflect its consideration.
Expressions such as "Muslim world" or "Middle East" have been replaced by more clarifying expressions. Others generalizations have also been taken into account and some modifications have been incorporated into the text. Thank you very much for your work, and for helping to improve this study.
Reviewer 2 Report
I enjoyed this discussion of Andalusian midwives from the 10th-14th centuries. I found the legal role that these women filled in society and inheritance particularly interesting and a novel addition to the field.
In revisions, I would like to see a more overt engagement with secondary sources from historical journals. I believe that if the scoping review could be run with Historical Abstracts as one of the databases it would ensure that the authors are engaging with historical scholarship which may or may not be indexed with medical/scientific databases like PubMed, Scopus, Web of Science, etc. In adding Historical Abstracts to the search, it might also be beneficial to expand the filters for this database to the last 20 years. Historical scholarship/publishing tends to move slower than that in scientific disciplines and what was published 20 years ago could still be very relevant to the conversation in which the authors are engaging.
The authors might also consider further connecting the activities of Andalusian midwives with the broader historiography of women’s care work in other areas of Europe at the time. This does not need to be a lengthy discussion (perhaps a paragraph or two) near the first discussion of historiography on page two. Women’s work, rarely featured in historical studies undertaken before the 1980s, a neglect that was especially apparent for women before the modern era. Therefore this article seeks to fill a void that scholars of other geographic areas are well familiar with, and it may be beneficial to illustrate these connections before going into the specifics about Andalusia.
Organizationally, the authors might want to consider moving up the primary source discussion of unknown women in medical treatises (section 3.2.3). The absence of women, and ‘reading against the grain’ in primary sources works better coming before a much longer discussion of where women are found in the primary documents.
I recommend that this article be revised and resubmitted for publication.

The text is readable throughout, and the meaning of words, phrases, etc., are clear. However, there are some sections of the text that could be reworded to make use of more active, argumentative language. For example, lines 131 to 135 could be broken up to have a shorter more argumentative topic sentence and then a second section with an explanation of the historiography. Maybe something like "Existing historiography has neglected care work and professionalization of Andalusian midwives." Then in the second sentence discuss what has been done in the historiography.
Author Response
Thank you for your review.
Following their recommendation, the search period has been corrected to 30 years, instead of 20, as it is more adapted to the search and the results obtained for this research. At the beginning, the search was limited to 10 years, but as explained in the text, this parameter was removed since works prior to this period had to be included due to their historical relevance for the research.
Because other reviewers have also not found clarity in the presentation of primary sources, as well as their scarcity, table 2 has been included, which specifies the specific primary sources that have been consulted and the places where they have been found.
Following your recommendation, a brief discussion is included, for comparison, on the historiography of women dedicated to care in other areas of Europe.
A complete revision of the text has been carried out in order to improve the quality of the translation into the English language.
Author Response
Thank you for your review.
The suggested revisions have been assessed and incorporated. The authors consider that they substantially improve the article.
P2:
- It is clarified in the text that we refer to medical science.
- It is specified that the medical figures are those described below. The lack of specification was a formatting problem caused by the translation.
- The difference between rational science and popular science is clarified, as well as the integration of both in some medical treatises and in the health practice of the time.
P3:
- Their academic training is clarified somewhat more, although we previously refer to a previous article published by this team that focuses on the health training of women in al-Andalus dedicated to care and their guardianship (Espina-Jerez et al, 2019). In any case, the sources do not clarify whether midwives, despite being able to read, were able to access medical treatises on gynaecology and obstetrics. However, it can be confirmed that women doctors could.
- Grammatical errors are corrected
P13: the misunderstanding about "a good raising" is clarified.
A complete revision of the text has been carried out in order to improve the quality of the translation into the English language.
Round 2
Reviewer 1 Report
Dear authors,
Thank you for the extensive revision of this article. In particular, the description of the primary sources consulted from the 9th - 14th centuries was a welcome and necessary addition. I think the greater attention to documenting and listing the primary sources is helpful.
Some of the language has been much improved, though I noticed that the world "slave" appears with little context, which I did not catch during my first read through. Perhaps the authors might want to include a note or context around that terminology.
I think the description of the qualitative approach is well articulated. In future, I would encourage the authors to engage more broadly with literature written by historians on the history of midwifery, which would enhance the analysis. The description of midwifery in medical and legal texts is a valuable contribution to the existing literature on childbirth in al-Andalus.
The authors have made substantial revisions which have enhanced the readability and argument of the paper.
Author Response
Dear Reviewer,
Thank you very much for your review.
Following your recommendation, it has been clarified in the cases where we refer to the female slave. One of the legal functions of the Andalusian midwive, or qābila, was to carry out a sanitary assessment of the state of health of the female slaves who were to be bought. Subsequently, the midwife could also be called upon to assess the state of health of the slave woman at your place of residence, once purchased.
Regarding the social framework of slavery in al-Ándalus, male and female, and its link with the health sciences, part of this group published a book chapter in which this subject was studied in depth: Espina-Jerez B, Alaminos MA, Gómez Cantarino S, Dominguez P, Moncunill E. Los cuidados brindados por la mujer a la esclavitud andalusí: X-XV centuries. In: Conocimientos, investigación y prácticas en el campo de la salud: innovación y cambio en competencias profesionales. Almería, ASUNIVEP, 2020, p. 455-9.
We are grateful for your appreciation of the clarification of the primary sources, and we will take into consideration your proposals for future improvement in the field of the history of care.